# PSO-LSSVR Assisted GPS/INS Positioning in Occlusion Region

**DOI:** 10.3390/s19235256

**Published:** 2019-11-29

**Authors:** Li Xiaoming, Tan Xinglong, Zhao Changsheng

**Affiliations:** 1Chuzhou College, School of Geographic Information and Tourism, ChuZhou 239000, China; 2AnHui Province Geographic Information Intelligent Perception and Service Engineering Laboratory, Chuzhou 239000, China; 3Jiangsu Normal University, School of Geographic Surveying and Mapping and Urban and Rural Planning, XuZhou 221000, China; tan_jsnu@126.com (T.X.); zhaocs1957@126.com (Z.C.)

**Keywords:** integrated navigation, least squares support vector regression, particle swarm optimization, fading adaptive filtering

## Abstract

Satellite signals are easily lost in complex observation environments and high dynamic motion states, and the position and posture errors of pure inertial navigation quickly diverges with time. This paper therefore proposes a scheme of occlusion region navigation based on least squares support vector regression (LSSVR), and particle swarm optimization (PSO), used to seek the global optimal parameters. Firstly, the scheme uses the incremental output of GPS (Global Positioning System) and Inertial Navigation System (INS) when the observation is normal as the training output and the training input sample, and then uses PSO to optimize the regression parameters of LSSVR. When the satellite signal is unavailable, the trained mapping model is used to predict the GPS pseudo position. Secondly, the observed anomaly is detected by the test statistic in the integrated navigation solution filtering estimation, and the exponential fading adaptive factor is introduced to suppress the influence of the abnormal pseudo observation value. The results indicate that the algorithm can predict the higher precision GPS position increment, and can effectively judge some abnormal observations that may occur in the predicted value, and adjust the observed noise covariance to suppress the anomaly observation, which can effectively improve the continuity and reliability of the integrated navigation system in the occlusion region.

## 1. Introduction

The multi-sensor integrated navigation system can obtain better positioning results when the observation environment is wide and the motion process is stable. However, satellite signals in forests, canyons, and tall buildings are subject to continuous or intermittent covered, multipath effects, and high-speed motion, causing the performance of integrated navigation systems to decline [1,2]. For the diagnosis of observed anomalies and dynamic model anomalies in dynamic navigation, most studies conduct hypothesis testing based on prediction residuals and state discrepancies, and then use the weighted iteration method to estimate the state robustly [3,4,5]. According to the influence of the gross error in the measurement vector on the state vector filter value, the robust Kalman filter model is deduced [6]. Guorong improved Sage adaptive filtering algorithm, which using UD (U is the unit upper triangular matrix, D is the diagonal matrix) decomposition to improves the adaptability of the system in high dynamic positioning [7]; Mohamed proposed moving window estimation and multi-model based adaptive estimation, which has a greater performance improvement than traditional filtering [8]. Tan proposed a method based on a genetic algorithm to optimize support vector regression parameters to assist abnormal fault detection due to the lack of redundant observations in dynamic navigation [9].

Machine learning algorithms have been extensively researched and developed in complex nonlinear problems. Intelligent algorithms such as LSSVR are based on the principle of structural risk minimization, which have global optimal solutions and high computational efficiency compared with neural network algorithms. There are no problems in curse of dimensionalityand topological structure that are difficult to determine, however, the selection of the radius parameter and the penalty parameter of the kernel function parameter are mostly empirical parameters, which will affect the accuracy of the regression [10]. Particle swarm optimization (PSO) is similar to a genetic algorithm. It searches for the optimal value according to the random solution given by the system, but the rules are simpler, with fast convergence and fewer setting parameters [11,12]. A nonlinear Gauss process regression (GPR) based on the PSO approach to perform vehicle position prediction during GPS outages was proposed which improved the position accuracy [13]. An optimal neural network-enhanced adaptive robust Kalman filter method has been proposed to improve the overall position accuracy during GNSS (Global Navigation Satellite System, GNSS) signal short-term outages [14]. Multi-sensor data fusion architecture with low cost has achieved a precision landing with less than 10 cm maximum estimated position error in GPS-denied areas [15].

This study combines LSSVR and the particle swarm optimization algorithm, by using the specific force increment and the angular rate increment of the inertial component output as the sample input of the nonlinear model training. The GPS position increment is used as the sample output, a nonlinear mapping model with optimal regression parameters is constructed in the global scope. In the case of the occlusion region, only the inertial output information is used to predict the reliable GPS pseudo-position. Then, the observed statistic is used to detect the abnormality of the observation, and the fading adaptive factor is taken to adjust the observed noise covariance matrix, suppressing the influence of abnormal observations, and assisting the GPS/INS integrated navigation system to maintain continuous and reliable positioning. This paper is divided into five sections. Section 2 describes the model of dynamics and observation in GPS/INS integrated navigation, and besides that, presents the determination method of fading adaptive factor. Section 3 introduces the steps of implementing the particle swarm optimization algorithm to optimize the parameters of least squares support vector machine regression. The verification of the proposed algorithm and analysis of the results follow in Section 4. The final section presents conclusions and discussions.

## 2. GPS/INS Adaptive Integrated Navigation Model

The 19 state parameters X considered in this article are expressed as follow:(1)$$X=[\delta p_{N},\delta p_{E},\delta p_{D},\delta v_{N},\delta v_{E},\delta v_{D},\delta \varphi_{N},\delta \varphi_{E},\delta \varphi_{D},\varepsilon_{N},\varepsilon_{E},\varepsilon_{D},\nabla_{N},\nabla_{E},\nabla_{D},\delta l_{N},\delta l_{E},\delta l_{D},\delta t{]}^{T}$$

Where $\delta p_{N},\delta p_{E},\delta p_{D}$ are the position errors, $\delta v_{N},\delta v_{E},\delta v_{D}$ are the velocity errors, $\delta \varphi_{N},\delta \varphi_{E},\delta \varphi_{D}$ are the attitude errors, $\varepsilon_{N},\varepsilon_{E},\varepsilon_{D}$ are the gyro drifts, $\nabla_{N},\nabla_{E},\nabla_{D}$ are the accelerometer biases, $\delta l_{N},\delta l_{E},\delta l_{D}$ represent the error of the lever arm measurement errors of the inertial navigation system to the GPS antenna, and δt is the time synchronization error.

The state space model is established in the “North-East-Down” coordinate system, the difference between the position of GPS and the inertial navigation as the observation input $Z_{k}$. The discretization equation of the GPS/INS loose combination model is written as:(2)$$\{\begin{array}{l} X_{k,k-1}=\varphi_{k-1,k-1}X_{k-1,k-1}+W_{k-1,k-1}\\ Z_{k}=H_{k}X_{k,k-1}+V_{k} \end{array}$$
where $\varphi_{k-1,k-1}$ is the state transition matrix at epoch *k-1*, $H_{k}$ is the observation matrix at epoch k; $W_{k-1,k-1}$, $V_{k}$ are the state noise vector and observed noise vector. The corresponding covariance matrices are written as $Q_{k-1,k-1}$ and $R_{k}$.

The integrated navigation system adopts the extended Kalman filter for optimal estimation [16,17]. The basic flow is written as follows:

State prediction:(3)$$\{\begin{array}{l} {\overset{\wedge }{X}}_{k,k-1}=\varphi_{k-1,k-1}X_{k-1,k-1}\\ P_{k,k-1}=\varphi_{k-1,k-1}P_{k-1}\varphi_{k-1,k-1}^{T}+Q_{k-1,k-1} \end{array}$$

State update:(4)$$\{\begin{array}{l} K_{k}=P_{k,k-1}H_{k}^{T}{\left[H_{k}P_{k,k-1}H_{k}+R_{k}\right]}^{-1}\\ P_{k}=\left[I-K_{k}H_{k}\right]P_{k,k-1}\\ \overset{\wedge }{X_{k}}={\overset{\wedge }{X}}_{k,k-1}+K_{k}\left(Z_{k}-H_{k}{\overset{\wedge }{X}}_{k,k-1}\right) \end{array}$$
where ${\overset{\wedge }{X}}_{k,k-1}$, $P_{k,k-1}$ are the state one-step estimate and state covariance matrix estimate; $K_{k}$ represents the Kalman matrix, which plays the role of making the state posterior estimation value closer to the true value; $\overset{\wedge }{X_{k}}$, $P_{k}$ is the state estimation value and its covariance matrix at epoch *k*. Through the iterative calculation of steps (3) and (4) to achieve the purpose of optimal estimation of the state and update [18,19].

### 2.1. Filter Anomaly Detection 

In the data processing of integrated navigation, the residual information based on the Kalman filtered estimate and the true value often cannot determine whether there is a gross error in the dynamic model and the observed model, which may cause the filter result to be unavailable. In this paper, it is considered that the observation information of the GPS output in the occlusion region scene must be abnormal. Under the premise that the dynamic model is accurately established, the prediction residual vector is used to construct the test statistic to determine whether there is an observation abnormality error.

The predicted residual $V_{k_{i}}$ is written as:(5)$$V_{k_{i}}=Z_{k_{i}}-H_{k_{i}}{\overset{\wedge }{X}}_{k,k-1}$$

The test statistic [Zhong et al., 2017] is expressed as:(6)$$\frac{V_{k_{i}}}{\delta_{V_{k}{}_{i}}}\sim N\left(0,1\right)$$
where δVki=(HkiPk,k−1HkiT+δi2)12, $H_{k_{i}}$ is the *i*-th row element of observation matrix at epoch *k*, $\delta_{i}^{2}$ represents the diagonal element of the observed noise covariance matrix. Assume the confidence level (1−α)%, the test statistic does not exceed the limit, it can be considered that there is no observation abnormality; if it exceeds the limit, the fading adaptive factor is introduced to adjust the system noise variance matrix, which achieves the purpose of identifying and suppressing the influence of observed abnormality.

### 2.2. Exponential Fading Adaptive Filter 

The variance $E\left(V_{k}V_{k}^{T}\right)$ of the prediction residual can be obtained from (5), and omitted the subscript *i*:(7)$$E\left(V_{k}V_{k}^{T}\right)=H_{k}P_{k,k-1}H_{k}^{T}+R_{k}$$

The variance of the prediction residual represents the lumped average of the random sequences, which can replace with time average in the discretization equation, moving the Equation (7), the observed noise variance matrix can be rewritten as:(8)$$R_{k}=\left(1-\frac{1}{k}\right)R_{k-1}+\frac{1}{k}\left(V_{k}V_{k}^{T}-H_{k}P_{k,k-1}H_{k}^{T}\right)$$

Replace the exponential fading factor b (0 < b < 1) in the above formula with $\frac{1}{k}$, order $\gamma_{k}=\frac{\gamma_{k-1}}{\gamma_{k-1}+b}$, Equation (8) can be rewritten as [20]:(9)$$R_{k}=\left(1-\gamma_{k}\right)R_{k-1}+\gamma_{k}\left(V_{k}V_{k}^{T}-H_{k}P_{k,k-1}H_{k}^{T}\right)$$

Considering the observed gross error is too large, the noise covariance determined by Equation (9) will increase the influence of the anomaly observation. Order $\beta_{k}=V_{k}V_{k}^{T}-H_{k}P_{k,k-1}H_{k}^{T}$, then the Equation (9) is rewritten as follows:(10)$$R_{k}^{i}=\{\begin{array}{c} \;\left(1-\gamma_{k}\right)R_{k-1}^{i}+\gamma_{k}R_{min}\;\;\;\;\;\;\;\;\;\;\;\;\;\;\;\;\;\;\;\;\;\;\;\;\;\;\;\beta_{k}^{i}<R_{min}\\ \left(1-\gamma_{k}\right)R_{k-1}^{i}+\gamma_{k}\left(V_{k}V_{k}^{T}-H_{k}P_{k,k-1}H_{k}^{T}\right)\;\;\;\;\;\;\;\;R_{min}<\beta_{k}^{i}<R_{max}\\ \;\;R_{max}\;\;\;\;\;\;\;\;\;\;\;\;\;\;\;\;\;\;\;\;\;\;\;\;\;\;\;\;\;\;\;\;\;\;\;\;\;\;\;\;\;\beta_{k}^{i}>R_{max}\; \end{array}$$

When the exponential fading adaptive factor is used to update the filter, the observation noise can be automatically adjusted according to the prediction residual, and the upper and lower limits of the noise variance are set to avoid inverse of matrix being negative and the filtering precision is reduced. At the same time, if the two adjacent differences of iterations do not exceed the limits, then stop the iteration.

## 3. PSO-LSSVR Assisted Positioning in Occlusion Region

The GPS/INS integrated navigation system will not obtain the satellite signals in the occlusion region, which results in the position of the receiver not being accurately obtained. The single inertial navigation will make the position and attitude error of the carrier accumulate rapidly with time without be corrected. However, LSSVR can construct the mapping model by using the specific force increment $\int_{t_{k-1}}^{t_{k}}f_{ib}^{b}dt$, the angular rate increment $\int_{t_{k-1}}^{t_{k}}\omega_{ib}^{b}dt$ of the inertial component and GPS three dimensional position increment $P_{t_{k}}$ when the integrated system is working normally [21,22,23], therfore PSO is used to search part of the global optimal parameters in the LSSVR algorithm. In the occlusion situation, the higher precision pseudo observations are predicted, and combine with the adaptive filtering to correct the INS error and estimate reliable navigation solution.

### 3.1. Least Squares Support Vector Regression Algorithm 

Support vector machine (SVM) is based on statistical learning theory and has the advantage of strong model generalization ability [24]. It is suitable for classification and regression of strong nonlinear problems. LSSVR is the least square form of SVM. The difference is that the former is equality constraints, the latter is inequality constraints. The main principle of LSSVR is to map the nonlinear samples that cannot be processed in the low-dimensional space to the high-dimensional feature space through the kernel function, so that the nonlinear samples can be linearly divided in this space to achieve the purpose of fitting prediction.

Assume the nonlinear training input sample $x_{i}$ includes: specific force increment and angular rate increment of the inertial component output, angle increment, training output sample $y_{i}$ is the coordinate increment of GPS, and the training data set D is written as:(11)$$D=\left\{\left(x_{1},y_{1}\right),\left(x_{2},y_{2}\right),\ldots ,\left(x_{n},y_{n}\right)\right\}$$

In this paper, the Gaussian kernel function $K\left(x,x^{\prime }\right)$ is used to map the sample data to the high-dimensional feature space, and then search for a optimal hyperplane, aim at the function value f(x) of the plane approximates the training output sample $y_{i}$:(12)$$\{\begin{array}{c} f\left(x\right)=\omega K\left(x,x^{\prime }\right)+b,i,j=1,2,\ldots ,n\\ K\left(x,x^{\prime }\right)=exp\left(-\frac{\left|\right|x-x^{\prime }{\left|\right|}^{2}}{2\delta^{2}}\right),\delta >0 \end{array}$$
where ω is the hyperplane normal vector of the sample point, using the L1 regularization solution can reduce the risk of overfitting and is easier to obtain the sparse solution; b is the bias top; δ is the bandwidth of the Gaussian kernel [25,26], when x is approximately equal to $x^{\prime }$, the kernel function value approximately to 1, this kernel function can map the original features to infinite dimensions.

The solution of hyperplane belongs to the solution of convex quadratic programming problem. Considering the existence of abnormal data outside the insensitive area, introducing the insensitive cost function ε, regularization parameters C and Lagrange multiplier $\alpha_{k}$ to construct the following function L, convert the above optimization problem to dual problem:(13)$$L\left(\omega ,b,\varepsilon ,\alpha \right)=\frac{1}{2}\omega^{T}\omega +\gamma \overset{N}{\underset{k=1}{\sum }}\varepsilon_{k}^{2}-\overset{N}{\underset{k=1}{\sum }}\alpha_{k}\left(\omega^{T}K\left(x_{k}\right)+b-y_{i}+\varepsilon_{k}\right)$$
where γ is the parameters of the kernel bandwidth in the kernel function, the parameters ω,b,ε,α in Equation (13) are solved for partial derivative to zero as the optimization condition, and a linear equation system for solving b,α is written as follows:(14)$$\left[\begin{array}{cccc} 0 & 1 & \ldots & 1\\ 1 & K\left(x_{1},x_{1}\right)+1/C & \ldots & K\left(x_{1},x_{n}\right)\\ \vdots & \vdots & \ddots & \vdots \\ 1 & K\left(x_{n},x_{1}\right) & \cdots & K\left(x_{n},x_{1}\right)+1/C \end{array}\right]\left[\begin{array}{c} b\\ \alpha_{1}\\ \vdots \\ \alpha_{n} \end{array}\right]=\left[\begin{array}{c} 0\\ y_{1}\\ \vdots \\ y_{t} \end{array}\right]$$

After solving the vector of b,α, the expression of the nonlinear prediction model can be written as:(15)$$f\left(x\right)=\overset{n}{\underset{i=1}{\sum }}\alpha_{i}K\left(x,x_{i}\right)+b$$

The performance of the least squares support vector machine is mainly determined by the kernel function parameter γ and the regularization parameter C. If the global optimal values of these two parameters can be searched, the learning and generalization ability of the mapping model will be greatly improved. In this paper, particle swarm optimization is used to optimize LSSVR.

### 3.2. Parameter Optimization Based on Particle Swarm Optimization Algorithm

Particle swarm optimization (PSO) is based on the idea that individual particles follow the optimal particles in the solution space at a certain speed in the population iteration [27]. Compared with other intelligent algorithms such as genetic algorithms, it has the advantages of faster convergence and fewer parameters that need to be set.

Assume that there are N particles in the D-dimensional space, their position is $X_{i}=\left\{X_{1},X_{2},\ldots ,X_{i}\right\}$, and the corresponding velocity is $V_{i}=\left\{V_{1},V_{2},\ldots ,V_{i}\right\}$, where i=1,2,…,n, Each $X_{i},V_{i}$ includes position and velocity information corresponding to two parameters (γ,C), the extremum of the *i*-th individual is marked as pbest, the population extremum is marked as gbest. In each iteration update calculation, the particle’s position and velocity update values are limited to $\left[X_{min},X_{max}\right]$ and $\left[V_{min},V_{max}\right]$, where the speed and position update formula is written as:(16)$$\{\begin{array}{l} V_{id}^{k}=wV_{id}^{k-1}+c_{1}r_{1}\left(pbest_{id}-X_{id}^{k-1}\right)+c_{2}r_{2}\left(gbest_{d}-X_{id}^{k-1}\right)\\ X_{id}^{k}=X_{id}^{k-1}+V_{id}^{k-1} \end{array}$$
where $V_{id}^{k}$ represents the *d*-th dimension component of the velocity vector of particle *i* at the *k*-th iteration; $X_{id}^{k}$ represents the d-th dimension component of the position vector of particle *i* at the *k*-th iteration; *c1* and *c2* represent acceleration constants for adjusting the maximum step size of learning; *r1* and *r2* represent random functions which values from 0 to 1, make the search random; w represents the inertia weight which used to adjust the search range of the solution space.

To calculate the fitness function value of each particle, we need to construct a suitable fitness function $f_{fitness}$. In this paper, we use LSSVR to predict the training data of the integrated navigation system based on the current position $\left(\gamma_{i},C_{i}\right)$ of each particle. The mean squared difference between the predicted position increment $y_{i}^{\prime }$ and the actual position increment $y_{i}$ is taken as the fitness function value:(17)$$f_{fitness}=\sqrt{\frac{\sum_{i=1}^{M}{\left(y_{i}^{\prime }-y_{i}\right)}^{2}}{M}}.$$

The steps of particle swarm optimization to optimize kernel function parameter and regularization parameter are as follows:Step 1:Initialize the population. The particle (γ,C) is randomly generated, which the range of the parameter is set to 0≤γ≤1000,0≤C≤100, the initialization speed of the parameter γ is set to [−500,500], and the initialization speed of the parameter C is set to [−50,50].Step 2:Calculating the fitness function value. The position of each particle is substituted into the LSSVR regression prediction, and the initial position is taken as the individual extremum position pbest of each particle, where in the position corresponding to the smallest fitness function value is taken as the global optimal position gbest of the particle group.Step 3:Particle update. Update the velocity and position of the particles according to Equation (16), and limit the particles that are not within the limited range of the velocity and position boundaries.Step 4:Substituting the position in Step 3 into Equation (17), calculating the fitness function value of each iteration of the particle, the current fitness function value is set to pbest and marking its position if the updated extreme value is smaller than the previous individual extreme value; if the current fitness value is better than the global extreme value, the current fitness value is set to gbest and marking its position, besides that, it is determined whether the absolute value of the global extreme value and the current fitness value difference satisfies the termination condition. Otherwise, iteratively continues until the condition is met, and then output the current global extremum and the global optimal value.

Substituting the obtained global optimal parameters (γ,C) into the regression model to calculate the position increments in three directions, complete the prediction of the pseudo position of the GPS receiver, and then integrate the conventional navigation position shift information to detect the anomaly of observations and adaptive filter estimation, after that obtain the output of combined navigation solution. Lastly, complete the correction of the inertial navigation error. The flow chart of the particle swarm optimization algorithm assisting LSSVR regression to predict GPS pseudo position is shown in Figure 1.

## 4. Results and Discussion

In this paper, the instruments used for data collection are two sets of Leica 1200 GPS receivers and one navigation level inertial measurement unit. Time of the total data is 1800 s, the GPS sampling period is 1 Hz, and the inertial measurement unit sampling period is 100 Hz. The accuracy of the differential GPS is 0.05 m, 0.05 m, 0.1 m in the three directions of the northeast, and the initial position error is set to 1 m, 1 m, 2 m, the initial attitude error set to 1°, 1°, 3°. The reference parameters of the gyroscope and accelerometer are shown in Table 1.

### 4.1. PSO-LSSVR Auxiliary Position Incremental Prediction

Select specific force increment, angular rate increment of inertial measurement units and GPS position increments between 450 and 600 s as training data to verify the prediction accuracy of the PSO-LSSVR algorithm. The motion state of the vehicle during this period mainly includes a straight line and a turn, and the speed is stable at 20 km/h. Assume that the satellite signal is lost after 900–1200 s, during which the motion state of the vehicle during this period mainly includes a straight line and three turns. The mapping model is used to predict the GPS position increment of the time. As shown in Figure 2, orange represents the training trajectory, green trajectory is predicted by the PSO-LSSVR algorithm during occlusion region, and black is the reference trajectory, solved by the GPS differential positioning solution.

The speed increment and angular increment of the carrier between 900 and 1200 s are shown in Figure 3. It is obvious that the speed of the vehicle movement process changes gently, and some epochs have observational outliers. The changes of pitch angle and roll angle are all within 0.002 radians, and the change in heading angle does not exceed 0.005 radians at the most. Besides that, the figure shows us the three stages of the curve motion of the vehicle, as indicated by the red circle mark.

We summarized the iterative optimization time for different population numbers as Table 2 represented, it is obvious that the iterative optimization time became longer with the population number increased. Figure 4 shows the prediction error of the position increments with four population numbers, when the population number is 10, the error of the position increments indicated minimum standard deviations, respectively were 0.134 m, 0.159 m, 0.120 m, and the prediction accuracy is the most stable. Comparing the mean of the position increments errors, the minimum mean is 0.213 m in the “D” direction where the population number is 10, and the mean errors in the “N” and ”E” direction are higher than the other populations, but there is no significant difference. In order to avoid affecting the real-time navigation, the optimization of the least square support vector regression model parameters in the global scope based on the particle swarm optimization algorithm can be determined according to the data collected in advance, and then directly used in navigation when GPS signal is unavailable.

The global optimization solution of the parameter of the PSO algorithm is shown in Table 3, where the population number is set to 10 and the iterative evolution number is 10. In order to compare the advantages and disadvantages of PSO–LSSVR algorithm, we added the least square support vector machine regression experiment based on the genetic algorithm, and the results are shown in the Figure 5. Figure 5a show the position increments of the training and predicted based on PSO–LSSVR and GA–LSSVR. The mean squared deviation of the training data in three directions are 0.051 m, 0.099 m, and 0.075 m respectively. The mean square error of PSO–LSSVR are 0.134 m, 0.159 m, and 0.120 m, the mean square error of GA–LSSVR are 0.233 m, 0.279 m, and 0.266 m. It is obvious that the accuracy of the predicted increment is on the sub-decimeter level. Moreover, it can be seen both algorithms have better prediction accuracy and PSO–LSSVR is more stable. There are some outliers in the results of GA–LSSVR algorithm, which may be related to the population size and mutation probability, under similar conditions, the optimization time of GA–LSSVR is about 101.273 s, which is longer than PSO–LSSVR.

Figure 5b shows the difference between the predicted increment and the reference increment. It can be more intuitively seen that the prediction results of most epochs are more consistent with the changes of the real trajectory of the vehicle. During the period of partial epoch, especially during the curve motion, the prediction increment accuracy is low, which may be due to the difference between the vehicle motion characteristics of the curve section and the curve section motion data in the training data during signal loss, which results in low prediction accuracy when using the model for mapping.

Figure 6a–c shows that the particle swarm optimization algorithm can achieve the global optimal fitness value after 3 to 5 iterations in “NED” directions, which indicates that it is feasible and effective to optimize the regression parameters of LSSVR by using PSO algorithm. Therefore, the scheme can predict the position increment of the GPS by inputting the output of the inertial measurement units during the occlusion region into the trained regression model, and then using adaptive filtering to suppress the abnormal observations that may occur in the prediction result. The finally-integrated navigation solution can correct the inertial system error more reliably after the estimate of the adaptive Kalman filter.

### 4.2. Fading Adaptive Filtering Based on PSO–LSSVR

When using PSO–LSSVR to predict the position increment during satellite signal loss, parts of epochs in the input data may have abnormal observations, which will cause abnormalities in the predicted position increments. These abnormalities will accumulate in subsequent observation epochs, and the reliability of the integrated navigation solution will be affected. Therefore, the test statistic is necessary for the detection of the fault information, and then calculate the integrated navigation solutions with EKF and AEKF (adaptive extend Kalman filter) two schemes which are based on the predicted values. The accuracy of navigation solutions is used to verify the effect of the adaptive filtering algorithm. The fault detection of the two schemes are shown in Figure 7:

In the observation model of this paper, the state degrees of freedom is 3, and the chi-square test thresholds corresponding to the significance level of 0.1 and 0.01 respectively are 6.251 and 11.345 from the table. The fault check level is taken as 0.1 here. If the chi-square test value of one single observation information is higher than the critical value, there is an abnormality, otherwise there is no abnormality. Figure 7a shows that the abnormality information of the observation model without the fading adaptive filtering correction can be detected by the test statistic, and the observation noise needs to be adjusted, and Figure 7b shows that after the adjustment of the adaptive factor, the reliability of the predicted observations can be improved.

Figure 8a–c shows the position error in the three directions of the “North-East-Down”, which use the EKF and the AEKF algorithm to calculate the integrated navigation solutions based on pseudo observations. It indicates that the accuracy of the navigation solution with the EKF algorithm is lower at several epochs, indicating that there is a large deviation in the predicted position increment of these epochs, however the position accuracy of the AEKF with the fading adaptive factor have been improved in three directions. The state error of inertial navigation is also well corrected. The error in the three-dimensional position solved by the AEKF algorithm is 1.831 m, 0.821 m and 1.846 m respectively; the error in the three-dimensional position solved by the EKF algorithm is 2.150 m, 1.248 m and 2.315 m. If positioned with dead reckoning (DR) only by INS, the positioning error will increase rapidly with time, thus indicating whether the accuracy of the combined navigation solution is reliable when the signal is lost or whether the system can maintain stability is also largely dependent on the prediction accuracy of the pseudo observations.

## 5. Conclusions

This paper aims that integrated navigation system in the occlusion region cannot work normally when the satellite signal is unavailable. The position error of the single inertial navigation cannot be corrected and accumulated over time. The PSO is used to optimize the partial regression parameters of LSSVR in the global scope. The optimal parameters are pre-trained while the system works normally. Once the signal is out of lock, reliable GPS pseudo-observation values can be predicted according to the output of the inertial measurement units, and then combined with the position of the dead reckoning to estimate the integrated navigation solution, compared with GA–LSSVR optimization algorithm, PSO–LSSVR algorithm has the advantages of higher regression accuracy and lower time consumption. Taking into account that the predicted observation value may exist abnormal observations, test statistics should be used to detect abnormal information, after that, introducing a fading adaptive factor, adaptively adjusting the observed noise covariance in the observed model, suppressing the influence of abnormal observations, and improving the filtering accuracy. Thereby enabling, when passing through the occlusion region for a certain period of time, the integrated navigation system can still have reliable, continuous and high-precision navigation and positioning, which has certain significance for engineering practice.

## Figures and Tables

**Figure 1 sensors-19-05256-f001:**
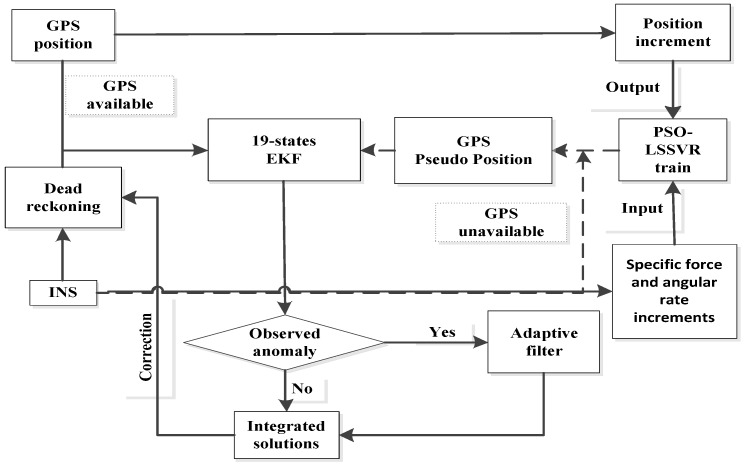
Primary swarm optimization–least squares support vector regression (PSO–LSSVR) auxiliary masking area positioning model.

**Figure 2 sensors-19-05256-f002:**
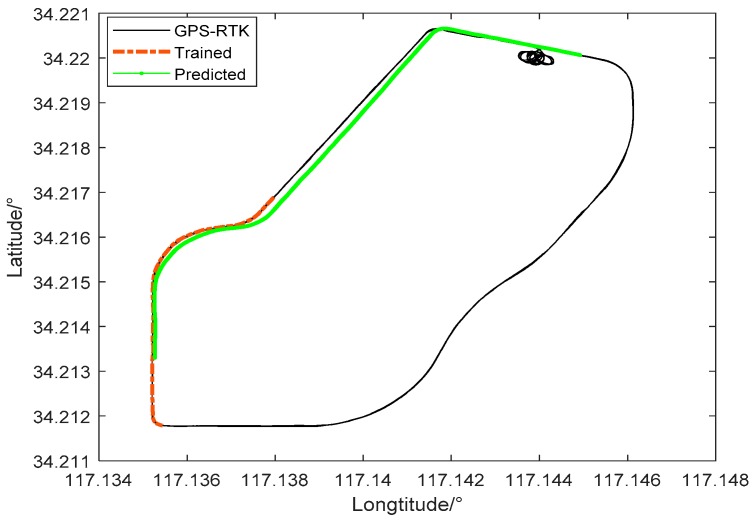
Predicted trajectory and real trajectory.

**Figure 3 sensors-19-05256-f003:**
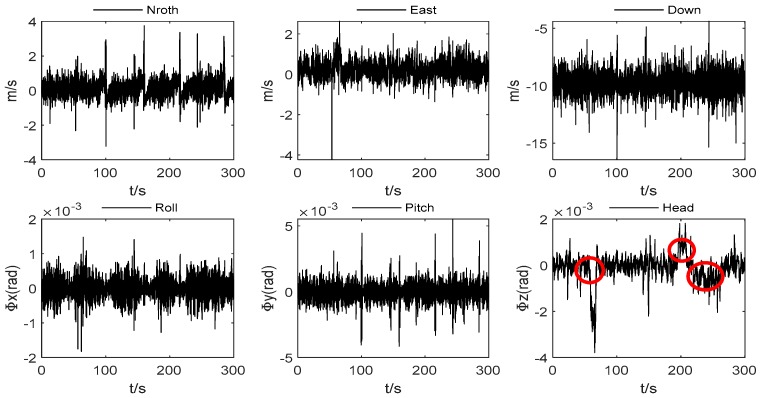
INS velocity increment and angle increment.

**Figure 4 sensors-19-05256-f004:**
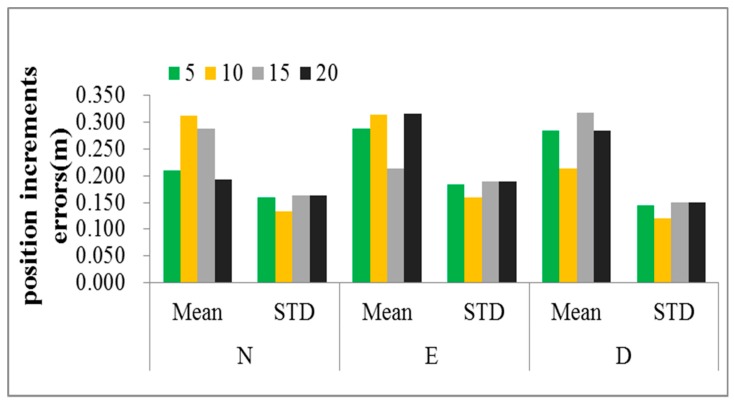
Comparison of the results of parameter optimization under different population numbers.

**Figure 5 sensors-19-05256-f005:**
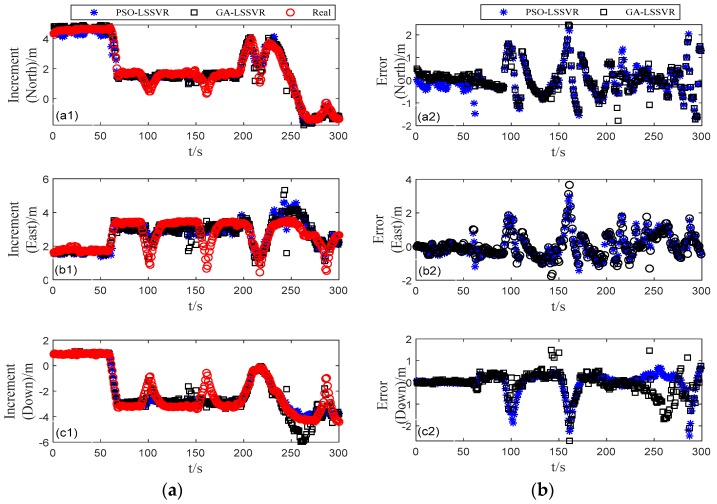
GPS position increment errors of different optimization algorithms: (**a**) the position increments predicted by different algorithms, (**b**) the errors of position increments predicted by different algorithms.

**Figure 6 sensors-19-05256-f006:**
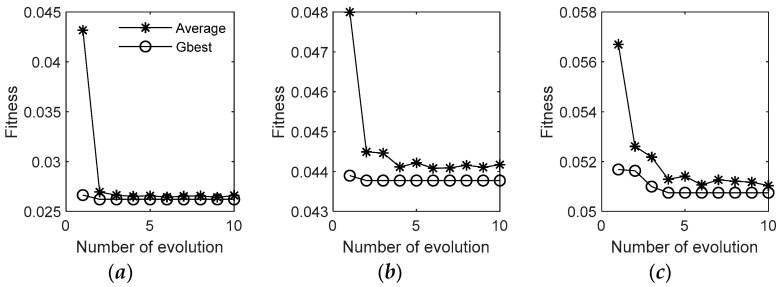
Fitness value iteration curve of particle swarm optimization algorithm: (**a**) represents the relationship between the value of the global optimal fitness function and the number of evolutions in the north, (**b**) represents the relationship between the value of the global optimal fitness function and the number of evolutions in the east, (**c**) represents the relationship between the value of the global optimal fitness function and the number of evolutions in the north in the down.

**Figure 7 sensors-19-05256-f007:**
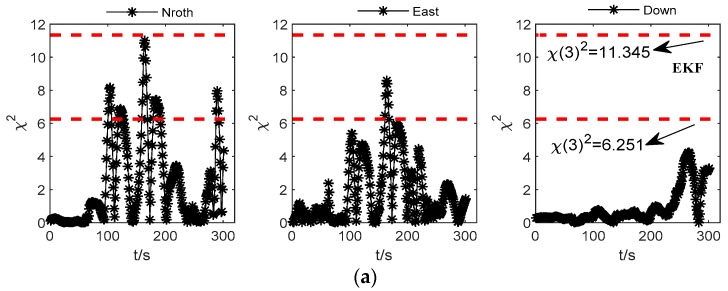
Comparison of (**a**) extended Kalman filter (EKF) and (**b**) adapted extended Kalman filter (AEKF) chi-square test values based on pseudo-observations.

**Figure 8 sensors-19-05256-f008:**
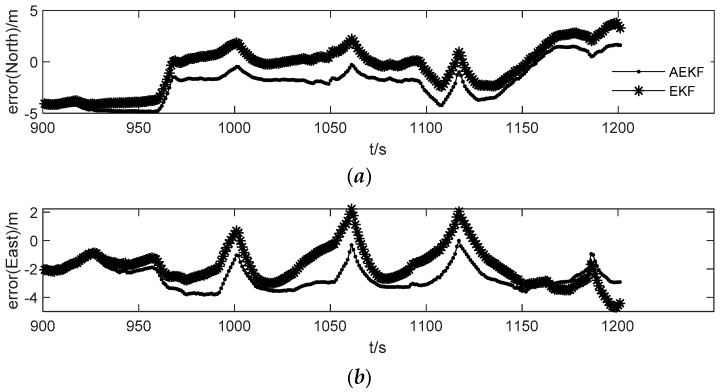
Comparison of AEKF and EKF integrated navigation results based on pseudo-observations: (**a**) represents the error of the integrated navigation position solution of AEKF and EKF in the north, (**b**) represents the error of the integrated navigation position solution of AEKF and EKF in the east. (**c**) represents the error of the integrated navigation position solution of AEKF and EKF in the donw.

**Table 1 sensors-19-05256-t001:** Main parameters of inertial sensor and receiver.

Parameters	Random Walk	Bias
Gyro	$0.067{}^{\circ}/\sqrt{hr}$	$20{}^{\circ}/\sqrt{hr}$
Accelerometer	$50\mu g/\sqrt{Hz}$	50 mg

**Table 2 sensors-19-05256-t002:** Time analysis of Particle Swarm Optimization under different population numbers.

	Number of Population			
	5	10	15	20
Iteration Time(s)	55.92	89.91	142.30	183.4

**Table 3 sensors-19-05256-t003:** Results of PSO-LSSVR.

Direction	PSO Parameter Optimization		RMS Error of Trained (m)	RMS Error of Predicted (m)
	*γ*	C		
Latitude	7.639	38.855	0.051	0.134
Longitude	957.511	96.524	0.099	0.159
Height	65.039	0.1	0.075	0.120

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
