# Peer review of "PSO-LSSVR Assisted GPS/INS Positioning in Occlusion Region"

_sensors, 2019, doi:10.3390/s19235256_

Round 1

Reviewer 1 Report

Lack of detailed up-to-date relative work analysis. The paper mentioned 19-dimensional state parameters were considered, but these parameters are not clearly presented in the subsequent derivation formula. In filter anomaly detection processing, what is the value of the confidence level Alpha or how to set the Alpha’s value? The iteration of machine learning and the calculation of multiple particles of particle filter are very time-consuming. The paper should analyze the sample number, particle number and related time complexity. Time index should also be given in the experiments. The experiments are not compared with other localization methods proposed to overcome gps failed problem in occlusion region, but only with different EKF methods.

Author Response

Due to the formula and picture in the reply cannot be displayed in the box. Please see the attachment.

Reviewer 2 Report

Even the well-known acronyms have to be defined: GPS, INS, UD?

Is line 66 correct: “attitude error?”

Lines 65-70 are unclear. Please rewrite them. 

Line 98: eq. (5), Please define N(01)

Can you write something related to calculus duration? The results contain the frequency, but not an estimation of the calculus on a specified processor.

Author Response

(The authors gave the same response as above.)
